# Impact of Stress and Dimension on Nanosheet Deformation during Channel Release of Gate-All-Around Device

**DOI:** 10.3390/mi14030611

**Published:** 2023-03-07

**Authors:** Jingwen Yang, Kun Chen, Dawei Wang, Tao Liu, Xin Sun, Qiang Wang, Ziqiang Huang, Zhecheng Pan, Saisheng Xu, Chen Wang, Chunlei Wu, Min Xu, David Wei Zhang

**Affiliations:** 1School of Microelectronics, Fudan University, Shanghai 200433, China; 2Shanghai Integrated Circuit Manufacturing Innovation Center Co., Ltd., Shanghai 201203, China

**Keywords:** Gate-All-Around (GAA), stress, nanosheet length, channel release, nanosheet deformation

## Abstract

In this paper, nanosheet deformation during channel release has been investigated and discussed in Gate-All-Around (GAA) transistors. Structures with different source/drain size and stacked Si nanosheet lengths were designed and fabricated. The experiment of channel release showed that the stress caused serious deformation to suspended nanosheets. With the guidance of the experiment result, based on simulation studies using the COMSOL Multiphysics and Sentaurus tools, it is confirmed that the stress applied on the channel from source/drain plays an important role in nanosheet deformation during the fabrication process. The deformation of Si nanosheets would cause a serious degradation of the device performance due to an inability to control the work function of the metal gate. This study proposed that the uniformly stacked GAA nanosheets structure could be successfully demonstrated with suitable channel stress engineering provided by fitting S/D size and an appropriate channel length. The conclusions provide useful guidelines for future stacked GAA transistors’ design and fabrication.

## 1. Introduction

Today, the whole complementary metal oxide semiconductor (CMOS) community is looking for solutions to complete the International Technology Roadmap for Semiconductors (ITRS) roadmap requirements. Maintaining a higher drive current with low off-state current leakage and controlling irresistible short channel effects (SCEs) are the primary challenges that limit the further downscaling of CMOS devices [1,2]. Gate-All-Around nanosheets transistors have become the most promising candidates for 3 nm node and beyond aggressive CMOS downscaling [3,4,5,6], owing to their superior electrostatics, better scalability, much stronger control over the gate electrical field and optimized power consumption compared to FinFET technology, overcoming the above-mentioned challenges [5,7,8].

In order to achieve a compatible fabrication approach with the mainstream FinFET process and improve the driving ability of the GAA device, several studies [9,10] have mentioned that stacked Si nanosheet GAA has been proposed with the use of SiGe as sacrificial layers to be removed selectively, versus a Si layer. Compared with the traditional bulk FinFET architecture, stacked Si nanosheet devices are fabricated thanks to the epitaxial growth of Si/SiGe multilayers, Si/SiGe highly selective etching and the conformal deposition of high-k dielectrics/metal gate (HKMG) stacks in between the Si nanosheets [11].

For the stacked GAA NS transistor, the channel release process is one of the main challenges in device fabrication [12], where nanosheet deformation such as stiction or collapse could happen if the process of forming suspended Si channels encountered a mechanical instability [13]. Usually, the suspended Si nanosheet deformation after channel release affects the thickness of the subsequent HKMG deposits, which results in a poorly adjustable work function. Conventionally, surface tension and capillary force on nanosheets during wet process have been reported as the main factor causing deformation [14,15]. Fortunately, this concern could be avoided by using critical point drying (CPD) and dry etching [16,17,18,19]. The other concern during the channel release is compressive stress associated with S/D, because the main transport orientation in GAA NS transistors changes from (110) to (100) [20], which exhibits higher electron mobility but lower hole mobility. To achieve N/P balance, channel stress engineering would be essential to introduce compressive stress in PMOS channel for enhancing the hole mobility [21]. The selective epitaxial growth of SiGe in the source and drain is considered one of the most effective methods for providing uniaxial compressive stress to the PMOS Si channel [22,23]. According to the mainstream GAA-FET fabrication process flow, the process of forming a suspended channel is executed after the epitaxial growth of the SiGe source and drain [19]. Therefore, the Si nanosheets are subjected to compressive stress from the source and drain during the process of channel release. So far, most research about the GAA device has focused on the optimization of electrical performances in terms of DC and AC. In this context, discussions on mechanical stability and implementing improvements seem timely in driving GAA development. 

In this work, the systematic investigation about the impact of S/D compressive stress on the mechanical stability of Si nanosheets during the channel release process is demonstrated. In particular, through extensive experiments, it has been found for the first time that the stress applied on nanosheet channels determines the deformation, which has been well explained and confirmed by insightful simulation based on Sentaurus SProcess and COMSOL Multiphysics. 

## 2. Stacked GAA Nanosheets Fabrication and Discussion

Figure 1a–d show the selected cross-section schematics of key steps in the GAA nanosheets transistor fabrication of the designed GAA transistors, with different Si nanosheet lengths and different S/D sizes being performed. Firstly, multilayer Si/SiGe (10 nm Si/10 nm Si_0.7_Ge_0.3_) superlattices were epitaxially grown on bulk Si (100) substrate. Then, the superlattice structure was patterned and etched down to the Si substrate by Inductive Coupled Plasma (ICP) to form a channel region as well as the S/D region. Moreover, the Si_0.7_Ge_0.3_ sacrificial layers were selectively removed through etching solutions including 10% HF, 30% H_2_O_2_ and 99.8% CH_3_COOH (mixing and aging for 72 h) with volume ratios of 1:2:3 [24,25,26,27]. After completing the suspended Si nanosheets using critical point drying, inter-layer (IL) formation and gate stack deposition of 3 nm HfO_2_ and 10 nm TiN by Atomic Layer Deposition (ALD) were completed immediately afterwards.

Two categories of experiments were designed for this work to study the correlation of device structure and stacking channel integrity after channel release. Table 1 lists the dimension parameters of the GAA device structure in Experiment A and Experiment B, respectively.

The fabricated fixed S/D width and length are 1 µm and 0.5 µm, while two different nanosheet dimensions with fixed width (30 nm [28]) and different lengths of 100 nm and 200 nm were used.The fabricated fixed nanosheet width Wch and length Lch are 30 nm and 100 nm, while two different S/D sizes with fixed width (1 µm) and different lengths of 2 µm and 0.5 µm were used.

The red box in Figure 2 represents the original position of the Si nanosheet. The offset of the actual positions of the three Si nanosheets from the red box represents the magnitude of their dislocation generation.

For experiment A, circled by the blue box in Figure 2a–d:

For a GAA device with 200 nm nanosheet length, as shown in Figure 2a,b, obvious sheets stiction occurred between nanosheets. It is not difficult to see that the bottom Si nanosheet undergoes the largest deformation, followed by the middle Si nanosheet, and the smallest deformation is in the top Si nanosheet. For the device with nanosheet length 100 nm, no visible nanosheet deformation occurred in three-layer stacked Si nanosheets, as observed in Figure 2d. Equally sized width and length of top, middle and bottom Si nanosheets was achieved, attributed to perfect fin etch and channel release control. This result indicates that in the case of providing the same volume S/D, the channel length has significant impact on the deformation of three-layer stacked Si nanosheets, especially the bottom nanosheet. 

For experiment B, circled by the red box in Figure 2c–f: 

Compared to the GAA device with smaller S/D size in Figure 2d, obvious sheets stiction occurred between nanosheets with larger S/D size. Therefore, in the case of the same Si nanosheet length, proper control of S/D size was employed to obtain an almost invariant Si nanosheet.

The experimental results clearly indicated that nanosheet length and S/D size are both the possible impact causing nanosheet deformation. In order to understand the impact of these key factors on the deformation of the Si nanosheet, insight TCAD simulations were demonstrated accordingly.

## 3. Simulation Results and Discussion

### 3.1. Stress Simulation of Si Nanosheet in Sentaurus 

Based on the above experiment parameters and process, a simplified 3D process simulation consistent with the steps in Figure 1 was carried out by employing the Sentaurus Process and Visual TCAD tools for obtaining the stress evolution of Si nanosheet. The purpose of simulation in Sentaurus is to verify that the stress is the main root cause for deformation. The structure parameters of a simulated three-layer stacked nanosheets device are depicted in Figure 3a, referring to the experiment structure in the previous section. The same nanosheets width, 30 nm, and Ge concentration of 30% in S/D and the sacrificial layer were adopted. Changing the trends of stress-ZZ in the Si nanosheet under three different steps, bulk wafer, fin patterning and channel release, was demonstrated in Figure 3b. Stress-ZZ subjected into Si_0.7_Ge_0.3_ is compressive stress (−2 GPa [29]) at the beginning because Si_0.7_Ge_0.3_ is grown according to the lattice epitaxy of Si, with no stress in the Si nanosheet. A point of compressive stress relaxation in Si_0.7_Ge_0.3_ transfers into Si after fin etch. Finally, Stress-ZZ subjected into a nanosheet changes from a little compressive stress to obvious compressive stress in subsequent channel release process steps, because of compressive stress in Si_0.7_Ge_0.3_ of S/D relaxation and its transmission into a Si nanosheet. The changing process of stress in the Si nanosheet, from no force to a large compressive stress, indicates that the compressive stress transmitted to the Si nanosheet by S/D may be one of the root causes for its displacement.

### 3.2. Mechanical Simulation in COMSOL

However, the Sentaurus TCAD tools cannot take the mechanical deformation in nanosheet into account, so we utilized the three-dimensional (3-D) numerical simulator COMSOL Multiphysics tool to study the mechanical displacement in GAA nanosheets by a finite element calculation approach because COMSOL includes a solid mechanics module. The simulation mesh size was defined as 1.5 nm. In addition, all simulations were performed under a steady-state condition.

Moreover, setting the exact, precise structure dimensions and material parameters in COMSOL tools are vital to assure simulation consistency; the details are as shown below:(1)Si nanosheets’ GAA structure as carried out in the COMSOL is exactly the same as in the experiment, including the same Si nanosheet dimension, same S/D dimension and same Ge component in S/D.(2)Especially, the dominant material parameter to determine the deformation of the Si nanosheet is Young’s modulus. Therefore, the setting of Young’s modulus of the material in COMSOL is completely consistent with the actual structure. Young’s modulus derived from the stiffness matrix is related to the stiffness factor describing the ability to resist deformation. Si and SiGe are both cubic crystal systems whose stiffness matrix is composed of three independent elastic constants, *C*_11_, *C*_12_ and *C*_44_, as shown:
(1)$$\left[\begin{array}{cccccc} C_{11} & C_{12} & C_{12} & 0 & 0 & 0\\ C_{12} & C_{11} & C_{12} & 0 & 0 & 0\\ C_{12} & C_{12} & C_{11} & 0 & 0 & 0\\ 0 & 0 & 0 & C_{44} & 0 & 0\\ 0 & 0 & 0 & 0 & C_{44} & 0\\ 0 & 0 & 0 & 0 & 0 & C_{44} \end{array}\right]$$

It is reported that the elastic constants *C*_11_, *C*_12_ and *C*_44_ of Si are separately 165.8 GPa, 63.9 GPa and 79.6 GPa. Similarly, elastic constants *C*_11_, *C*_12_ and *C*_44_ of Ge are separately 128.5 GPa, 48.3 GPa and 66.8 GPa. Therefore, the elastic constant of Si_0.7_Ge_0.3_ [30] used in the source/drain simulation could be calculated by Formulas (2)–(4), where *x* = 0.3.
(2)$$C_{11}=\left(165.8-37.3x\right)\;\mathrm{GPa}\;300\;K$$
(3)$$C_{12}=\left(63.9-15.6x\right)\;\mathrm{GPa}\;300\;K$$
(4)$$C_{44}=\left(79.6-12.8x\right)\;\mathrm{GPa}\;300\;K$$

Then, put the elastic constants of Si and Si_0.7_Ge_0.3_ into (5)–(8) to obtain Young’s modulus [31]. Finally, the Young’s modulus of Si and Si_0.7_Ge_0.3_ were both put into material properties of the COMSOL simulation separately.
(5)$$\mathrm{Voigt}\;\mathrm{average}:\;G_{V}=\frac{\left[C_{11}-C_{12}+3C_{44}\right]}{5}\;B_{V}=\frac{C_{11}+2C_{12}}{3}$$
(6)$$\mathrm{Reuss}\;\mathrm{average}:\;G_{R}=\frac{5C_{44}\left(C_{11}-C_{12}\right)}{3\left(C_{11}-C_{12}\right)+C_{44}}\;B_{V}=\frac{C_{11}+2C_{12}}{3}$$
(7)$$\mathrm{Hill}\;\mathrm{average}:\;G=\frac{G_{V}+G_{R}}{2}\;B=\frac{B_{V}+B_{R}}{2}$$
(8)$$\mathrm{Young}’s\;\mathrm{modulus}:\;E=\frac{9GB}{3B+G}$$

Compressive stress, which is the most significant parameter in this mechanical displacement simulation, was applied to the SiGe S/D. After setting the material parameters, the compressive stress in fully strained S/D, −2 GPa, was put into the device in COMSOL the same as the structure in the experiment. Finally, the deformation of the suspended Si nanosheet was obtained in the COMSOL mechanical simulation. The process of deformation simulation is the result of the natural relaxation of stress −2 GPa during channel release. Therefore, for simulation in COMSOL, the input condition is the SiGe S/D compressive stress, and the simulation result is the deformation of the Si nanosheet. To verify the realism of the COMSOL simulation deformation, the stress distribution after fin etch was simulated in both COMSOL and Sentaurus TCAD, adopting the same SiGe S/D stress. The stress mappings are found to be very similar, so it can be guaranteed that the model of COMSOL simulated deformation is feasible.

### 3.3. Result Discussion

#### 3.3.1. Impact of Channel Length

GAA nanosheets devices with different nanosheet lengths of 100 nm and 200 nm have been simulated. Additionally, a fixed S/D size (width: 1 µm; length: 0.5 µm) and nanosheet width of 30 nm were used. From the comparison of Figure 4a,b, with the same compressive stress −2 GPa provided by S/D, the deformation of the nanosheet of length 200 nm was much more severe than 100 nm. As explained by the stiffness factor, whose expression is EA/L (“E” represents Young’s modulus of Si nanosheet, “A” represents cross-sectional area and “L” is length of Si nanosheet), when it is subjected to a force it increases with the length of the nanosheet decreasing. An obvious trend could be observed in the figures: the nanosheets displacement values increase for longer nanosheets, consistent with the conclusion obtained in Experiment A. Moreover, the three-layer stacked Si nanosheets in both Experiment A and the simulation have the largest displacement in the bottom layer, validating the reliability of the displacement simulation. In addition, the displacement of the bottom nanosheet in Figure 4a,b along the Si nanosheet direction has been plotted in Figure 4c.

#### 3.3.2. Impact of Pad Length

The S/D regions with varied length from 0.2 µm to 2.0 µm and a fixed width of 1 µm were used, while the same nanosheets width of 30 nm and length 100 nm were adopted. Figure 5a displays the comparison of stress in the bottom sheets calculated by Sentaurus and COMSOL. The stress calculated by Sentaurus represents the undeformed stress in the bottom nanosheet, while the stress calculated by COMSOL represents the post-deformation stress in the bottom nanosheet. The two kinds of channel stress both increase with increasing S/D pad length, match with the trend of S/D stress and tend to saturate beyond the pad length of 1 µm. Moreover, the variation between undeformed and post-deformation stress-ZZ is so obvious with increasing S/D region length that the problem of stress-ZZ loss becomes non-negligible due to the deformation of the Si nanosheet.

Similarly, according to the structural parameters of Experiment B, the compressive stress value −2 GPa was input into the GAA device with different S/D lengths in Comsol, and the corresponding deformation was obtained by COMSOL simulation, as shown in schematic Figure 5b,c. In addition, significant bottom nanosheet deformation has been observed in COMSOL simulations for larger S/D length compared with smaller S/D length, coinciding with the conclusion of Experiment B, as shown in Figure 5d. Meanwhile, Figure 5a shows the results more completely; that the deformation calculated in COMSOL increases with the increase of stress delivered by S/D. All the evidence suggests that stress provided by S/D would be a key factor causing nanosheets bending or even collapse.

The simulation results agreed excellently with the aforementioned experiment analysis and strongly verified that nanosheet length and stress-ZZ provided by S/D are both the key origins causing nanosheet deformation. Therefore, deliberating S/D stress engineering and adopting special process engineering to tackle displacement problems for different lengths of nanosheets during the device fabrication are vital to achieve uniform nanosheets structure. However, for extending to the realistic GAA device structure with epitaxial S/D, especially PMOS with SiGe S/D, because the main transport orientation in the GAA NS Nanosheet transistors changes from (110) to (100), which exhibits higher electron mobility but lower hole mobility, the channel stress engineering plays more important roles, especially for PMOS performance, when achieving N/P balance [22]. Giving the nanosheet as much stress as possible while ensuring no obvious collapse plays a critical role in the GAA device fabrication.

## 4. Conclusions

The Si nanosheet deformation will not only have a bad effect on the work function of the metal gate but will also cause stress loss in the channel, so in this work the impact factors of Si nanosheet deformation have been investigated and discussed based on both experiment and TCAD simulation results for the first time. Three-layer stacked GAA nanosheets devices under different strain conditions and Si nanosheet length have been designed and successfully fabricated. Furthermore, Sentaurus and Comsol simulations have been carried out to validate the impact of stress provided by S/D and Si nanosheets’ length on its deformation. The results of the simulation synchronized with experiments suggest that the two factors both play an important role in Si nanosheet deformation. This paper highlighted that deliberating strain engineering and employing suitable nanosheet dimensions in device fabrication are vital to achieve highly uniform nanosheets as well as uniform inter-channel space. Additionally, obtaining greater stress-ZZ in silicon nanosheets without affecting the subsequent steps of the HKMG could be of great concern in extremely scaled GAA nanosheets transistors study. The conclusion will be helpful for subsequent stacked GAA transistors fabrication.

## Figures and Tables

**Figure 1 micromachines-14-00611-f001:**
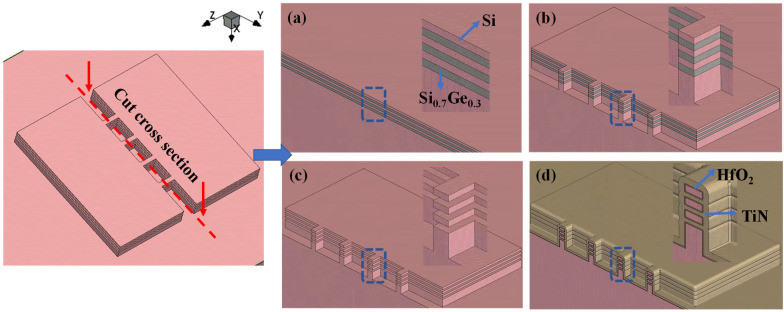
Cross-section of key fabrication steps of GAA Si nanosheet fabrication: (**a**) Multilayer epitaxy of Si/SiGe superlattice. (**b**) S/D and channel patterning. (**c**) Channel release of Si nanosheets. (**d**) HKMG deposition.

**Figure 2 micromachines-14-00611-f002:**
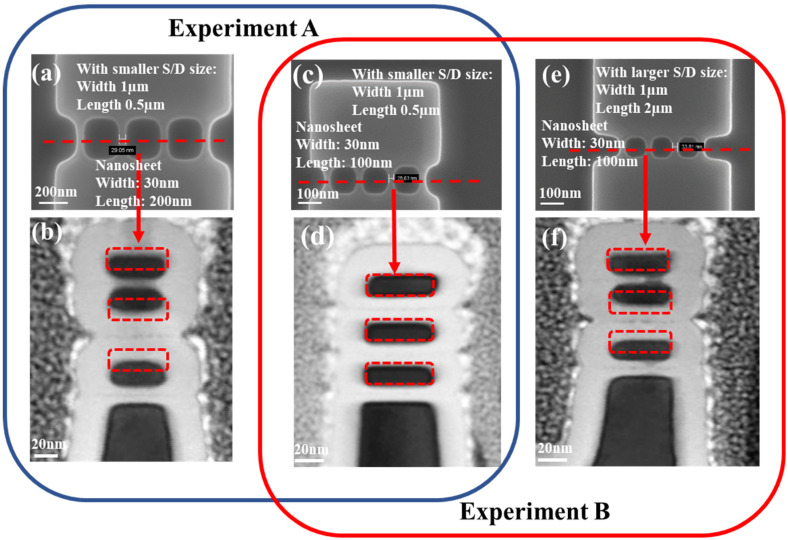
(**a**) Top view SEM image and (**b**) cross-section view TEM image of 3-layer GAA Si nanosheets with length 200 nm and smaller S/D size; (**c**) Top view SEM image and (**d**) cross-section view TEM image of 3-layer GAA Si nanosheets with length 100 nm and smaller S/D size. (**e**) Top view SEM image and (**f**) cross-section view TEM image of 3-layer GAA Si nanosheets with length 100 nm and larger S/D size.

**Figure 3 micromachines-14-00611-f003:**
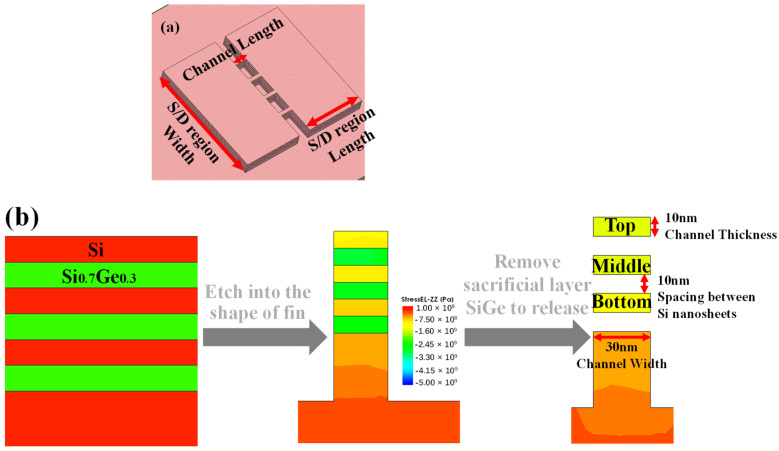
(**a**) The top view of the simulated three-layer stacked nanosheets transistor schematic; (**b**) cross-sectional view of stress distribution under different process steps in Sentaurus.

**Figure 4 micromachines-14-00611-f004:**
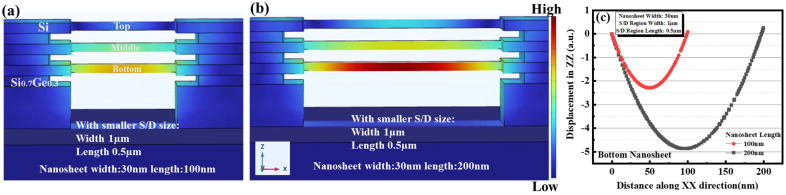
Different GAA Si nanosheet lengths (**a**) 100 nm and (**b**) 200 nm with smaller S/D size schematics of displacement mapping of the top, middle and bottom nanosheet in COMSOL (Si nanosheet width: 30 nm). (**c**) Extracted displacement of the bottom nanosheet in (**b**) along XX direction.

**Figure 5 micromachines-14-00611-f005:**
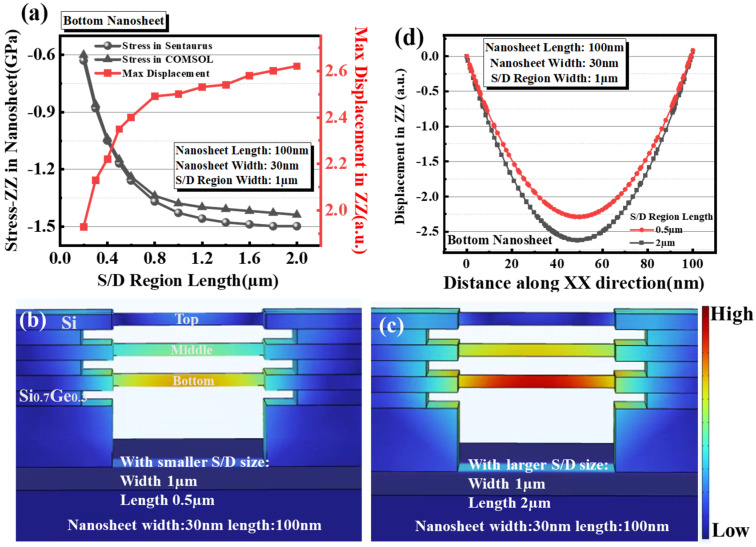
(**a**) The stress values calculated by Sentaurus and COMSOL and displacement in bottom nanosheet channel as a function of varied S/D region length. GAA device with different S/D length (**b**) 0.5 µm and (**c**) 2 µm schematics of displacement mapping of top, middle and bottom nanosheet in COMSOL (Si nanosheet length: 100 nm width: 30 nm). (**d**) Extracted displacement of bottom nanosheet along XX direction with S/D region length 0.5 µm and 2 µm.

**Table 1 micromachines-14-00611-t001:** Physical parameters of the GAA device in experiment.

Experiment	S/D Width	S/D Length	Nanosheet Length	Nanosheet Width	Nanosheet Thickness
A	1 µm	0.5 µm	100 nm	30 nm	10 nm
			200 nm		
B	1 µm	2 µm	100 nm	30 nm	10 nm
		0.5 µm			

## Data Availability

The data that support the findings of this study are available from the corresponding author upon reasonable request.

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
