# Peer review of "Impact of Stress and Dimension on Nanosheet Deformation during Channel Release of Gate-All-Around Device"

_micromachines, 2023, doi:10.3390/mi14030611_

Round 1

Reviewer 1 Report

The author discusses the deformation of nanosheets during channel release in gate-all-around (GAA) nanosheet transistors. In this work, uniformly stacked GAA nanosheet structures have been demonstrated with an appropriate channel stress engineering approach provided by the fitting of S/D size and appropriate channel length. This topic is of great interest to me as it pertains to GAA NS MOSFETs. It seems, the authors extended their work from a conference paper entitled "Investigation of nanosheet deformation during channel release in gate-all-around nanosheet transistors, 2022 China semiconductor technology international conference (CSTIC)".

1. This paper first experimentally and computationally explore the suitable conditions for channel release. It is helpful to optimize the stress conditions for process simulations applied to future stacked GAA devices. However, the paper title should be prepared precisely in the revised version.

2. It is necessary for the author to provide a more detailed description of the properties of GAA NS transistors in the section I, so please elaborate on the introduction. Also, the computational scheme is unclear. The authors only mentioned about the elastic constant and Young's modulus of SiGe. However, the channels are made of silicon, as shown in Fig. 1(d).

3. It is not clearly mentioning the properties of structure, draw a table listing the physical parameters of the device.

4. In section II, the authors describe the automatic layer deposition (ALD) method for the deposition of 3nm HfO2 and 10nm TiN gate stacks. Originally, in Si and HfO2, the SiO2 interfacial layer served as a barrier that prevented the diffusion of Hf atoms into the Si channel from the gate stack. This can result in altered electrical characteristics, leading to defects that may adversely affect the performance of the device. Why do authors use only HfO2 between the Si channel and TiN gate metal?

5. There are repetitions in the writing of abstract and conclusion of the paper. This should be re-polished entirely. For example, it is better to quantify the improvement or benefit of the proposed technique in Abstract. The degradation of stability of the device characteristics due to the channel deformation should be indicated to enhance the main contribution of this work.

6. The settings between SProcess and COMSOL should be provided in detail.

7. Furthermore, it is better to compare the fabricated results with Fig. 5 which shows that a severe deformation will occur in the bottom channel. Figs. 5 and 6 should compare with experimental results in order to clarify the difference between measurement and simulation. This accuracy verification will confirm the asserted arguments.

8. There are few reference papers on channel release and stress. This part should be polished.

9. In Fig. 2, the photo is not clear, and the channel deformation cannot be seen. And there is no SEM bar marked. It is not obvious that the nanosheet stiction occurs in the structure with a longer channel Fig. 2(a) while Fig. 2(b) is without the nanosheet deformation. It should be exhibited in the cross-section plot.

10. In Fig. 3(b), Red box offset. Equation has a wrong number. For example, (5). Typo errors, fonts and other formats are inconsistent. Poor writing in English. Grammar and academic terminology should be re-edited carefully. Format of references should be unified.

Reviewer 3 Report

This manuscript uses finite element analysis to model the strain-stress relationship in a gate-all-around nanosheet transistor. The hypothesis that the stress is leading to the deformation seem to be innovative, and the results are clearly presented. However, there are a few issues that need to be addressed before this manuscript can be considered for publication.

1.       The introduction should include more background information on the following aspects with proper citations:

a.       Brief description of channel release process

b.       More recent literature to demonstrate that the nanosheet deformation is a challenge in the channel release process

c.       Current approaches to reduce nanosheet deformation

2.       It seems that the configuration can be approximated to a beam bending problem with analytical solutions. The authors should consider comparing the modeling results to the analytical solution.

3.       In section 3.1, “the input condition of the simulation is the SiGe S/D compressive stress in the fin etch, and the simulation result is the deformation of the Si nanosheet.” (line 104-105). What is the value of SiGe S/D compressive stress? Reference also needs to be included.

4.       The authors should include more discussion why they believe the stress is the main root cause for deformation.  

Round 2

Reviewer 1 Report

The authors answered the reviewer's comments and sugestions properly. However, the writing should be carefully and entirely polished in English before the final acceptance.

Reviewer 3 Report

The authors has improved on the introduction and included sufficient details on the simulation setup. I recommend its publication in Micromachines.